# PHASE2VEC:
# DYNAMICAL SYSTEMS EMBEDDING WITH A PHYSICS-INFORMED CONVOLUTIONAL NETWORK

**Matthew Ricci**
School of Computer Science and Engineering
The Hebrew University
Jerusalem, Israel
matthew.ricci@mail.huji.ac.il

**Noa Moriel**
School of Computer Science and Engineering
The Hebrew University
Jerusalem, Israel
noa.moriel@mail.huji.ac.il

**Zoe Piran**
School of Computer Science and Engineering
The Hebrew University
Jerusalem, Israel
zoe.piran@mail.huji.ac.il

**Mor Nitzan**
School of Computer Science and Engineering,
Racah Institute of Physics, Faculty of Medicine
The Hebrew University
Jerusalem, Israel
mor.nitzan@mail.huji.ac.il

## ABSTRACT

Dynamical systems are found in innumerable forms across the physical and biological sciences, yet all these systems fall naturally into equivalence classes: conservative or dissipative, stable or unstable, compressible or incompressible. Predicting these classes from data remains an essential open challenge in computational physics on which existing time-series classification methods struggle. Here, we propose, phase2vec, an embedding method that learns high-quality, physically-meaningful representations of low-dimensional dynamical systems without supervision. Our embeddings are produced by a convolutional backbone that extracts geometric features from flow data and minimizes a physically-informed vector field reconstruction loss. The trained architecture can not only predict the equations of unseen data, but also produces embeddings that encode meaningful physical properties of input data (e.g. stability of fixed points, conservation of energy, and the incompressibility of flows) more faithfully than standard blackbox classifiers and state-of-the-art time series classification techniques. We additionally apply our embeddings to the analysis of meteorological data, showing we can detect climatically meaningful features. Collectively, our results demonstrate the viability of embedding approaches for the discovery of dynamical features in physical systems.

## 1 INTRODUCTION

The application of deep neural networks to the prediction (Lusch et al., 2018), control (Haluszczynski & Räth, 2021), and basic understanding (Raissi et al., 2019) of dynamical systems has spurred important advances across the sciences, from neuroscience (Sussillo et al., 2016) to physics (Karniadakis et al., 2021), from earth and climate science (Reichstein et al., 2019; Fresca et al., 2020) to computational biology (Sapoval et al., 2022). However, while deep learning has already contributed significantly to the analysis of single systems, its role in understanding the underlying principles of dynamical systems *in general* remains somewhat limited. The ability to predict or control one system, after all, tells us little about how to do so with another. Crucially, without the ability to generalize across behaviors of numerous dynamical systems, the problem of constructing new systems is quite difficult, and most current attempts rely on exhaustive search through system parameters (Hart et al., 2012; Scholes et al., 2019).

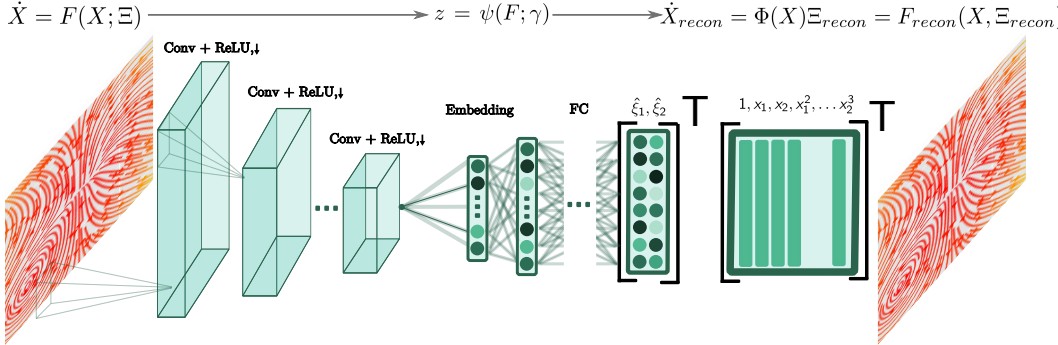

$$\dot{X} = F(X; \Xi) \longrightarrow z = \psi(F; \gamma) \longrightarrow \dot{X}_{recon} = \Phi(X)\Xi_{recon} = F_{recon}(X, \Xi_{recon})$$

Figure 1: *phase2vec pipeline*. phase2vec learns high-quality embeddings of dynamical systems from phase space data. An input vector field (represented as a stream plot, i.e. a set of trajectories) is passed through a series of convolutional, rectification, and downsampling layers. Terminal convolutional features are aggregated in a relatively low-dimensional ($d = 100$) layer before being mapped to a set of estimated coefficients, $\Xi_{recon}$. These coefficients are used to weight a dictionary of polynomials in $q$ variables (depicted $q = 2$), and the resulting linear combination (depicted transposed above) comprises the estimated governing equation, $F_{recon}(X, \Xi_{recon})$ associated to the reconstructed vector field.

A machine learning framework that could elucidate the latent dynamical structure across numerous systems having different governing equations and parameters would be an important first step in this direction. However, progress on this front has been limited, largely because representations of even a single dynamical system are necessarily high-dimensional (many initial conditions over time) and the problem of understanding single systems in detail is already as daunting as it is useful. The few existing approaches that analyze multiple systems simultaneously focus on the limited setting of rapid adaptation to new parameter regimes for governing equations (Kirchmeyer et al., 2022) or rely on hardwired features and laborious search over model manifolds (Quinn et al., 2021).

To address this challenge, we propose phase2vec, a dynamical systems embedding model which learns high-quality, low-dimensional, physically-meaningful representations of dynamical systems (Fig. 1). We focus here on two- and three-dimensional dynamics because of their preponderance in nature, ranging from classical non-linear models of population growth like the Lotka-Volterra model to increasingly important models of climate dynamics (for applications to both, see Sec. 4). Similarly to many word embedding approaches (Mikolov et al., 2013), which approximate the meaning of words using statistical regularity instead of formal semantics, our dynamical embeddings seek to model the "semantics" of dynamical systems from data instead of via analytical investigation. phase2vec is so-called since it is based on a vector of convolutional features extracted from the vector field representing the dynamics in phase space of input data. To encourage physically-meaningful solutions from our encodings, we use a decoder which reconstructs input dynamics from a library of basis functions. The full pipeline is trained with a reconstruction loss placing special emphasis on fixed points, giving phase2vec a physically-informed "learning bias" (following the terminology for principles of physics-informed learning suggested in (Karniadakis et al., 2021), Box 2).

Like other embedding methods, we train our system on an auxiliary task, in our case one involving equation reconstruction. We show that this auxiliary task, combined with a physics-informed loss, steers the network towards learning dynamically stable, informative embeddings[1].

The main contributions of this paper are:

1. phase2vec, a novel neural-network-based embedding method that can be used to learn low-dimensional representations of dynamical systems in an unsupervised manner.

2. A demonstration that these embeddings can be used to decode the governing equations of testing data, recover sparse underlying models, and denoise input data in a way that pre-

---

[1]Code available here: https://github.com/nitzanlab/phase2vec

serves dynamical characteristics better than per-equation fitting approaches (e.g. LASSO; (Tibshirani, 1996)), even on noisy data.

3. Quantitative analysis showing that important physical properties (conservation of energy, incompressibility, class of dynamical stability) can be decoded from `phase2vec` embeddings with much greater accuracy than both blackbox classifiers and contemporary time series classification approaches.

4. An application of `phase2vec` to the analysis of climate data, demonstrating that the embeddings capture meaningful structure in real data, such as the characteristics of distinct temperature zones.

## 2 RELATED WORK

**Data-driven discovery of governing equations**    There is a large body of work concerning the discovery of governing equations of single systems from time series data (reviewed in (Timme & Casadiego, 2014)). Early work by (Bongard & Lipson, 2007; Schmidt & Lipson, 2009) fit the coefficients of an explicit basis of polynomials to time series data using least squares, an approach which was modernized in the sparse reconstruction method of SINDy (Brunton et al., 2016) and its autoencoder formulation in (Champion et al., 2019). Other approaches eschew the explicit modeling of dynamical equations in favor of implicitly representing the dynamical transition as the discrete-time action of an autoencoder on state samples. For example, (Lusch et al., 2018) approximated the action of the Koopman operator in such a set-up, while (Bramburger et al., 2021) used an autoencoder to learn topologically conjugate representations of unseen equations. Importantly, all of the above methods focus on a specific system at hand and do not generalize to additional settings.

**Generalization across physical systems**    While there has been some progress in estimating equations for single systems, inferring the dynamics of multiple systems simultaneously has proven more challenging (Brown & Sethna, 2003). The manifold boundary approximation method of (Quinn et al., 2021) has been used in multiple cases (Transtrum & Qiu, 2014) to estimate the minimal mechanisms underlying behavioral archetypes within a single parameterized dynamical family (e.g. a family of Ising models parameterized by coupling strength (Teoh et al., 2020)). This method can be used to identify dynamics that happen to lie on the boundaries of the model manifold, but its use in identifying arbitrary, user-defined dynamical classes is less straightforward. A more recent line of work (Clavera et al., 2019; Lee et al., 2020; Yin et al., 2021; Kirchmeyer et al., 2022; Wang et al., 2022) has taken an adaptation approach whereby the representation of dynamics observed during a training period can be quickly updated to a new setting in which the parameters of the underlying system have changed. Here, too, the focus is the generalization to new parameter regimes of the same underlying equation. To our knowledge, the problem of learning generalizable representations of dynamical models whose functional forms vary widely has not been systematically addressed.

**Vector-field-based approaches**    All of the above approaches have taken time series as their input data. Instead, vector field representations can be generated from known governing equations or from time series measurements, for instance by binning velocities (Sneed & Komaee, 2021) or by fitting a statistical model (e.g. (Qiu et al., 2022)). Among the approaches which rely on vector field data is (Battistelli & Tesi, 2020), which provided theoretical guarantees for the performance of two-class vector field classifiers for linear systems. (Ye et al., 2020) used a supervised convolutional network to analyze a vector field arising from aerodynamic simulations, but they neither addressed the problem of generalization across systems or equations nor the problem of classifying general physical properties. Others sought to construct vector fields subject to the constraints on the number, location, and stability of fixed points which are mathematically imposed by Conley index theory(Conley, 1978; Zhang et al., 2006; Chen et al., 2008). Our approach not only combines the classifier method of (Battistelli & Tesi, 2020) with the physical biases of (Zhang et al., 2006; Chen et al., 2008), but importantly introduces deep-learning-based feature extraction, supporting generalization across complex non-linear dynamics.

## 3 METHODOLOGY

Let $\mathcal{D} = \{F_i\}_{i=1}^m$ be a set of continuously differentiable, parameterized vector fields on $\mathbb{R}^q$, each with parameter vector $\Xi_i$. Each $F \in \mathcal{D}$ is associated to a dynamical system via

$$\dot{X} = F(X; \Xi). \tag{1}$$

When $F$ is identified with a dynamical system in this manner, we refer to $\mathbb{R}^q$ as the system's *phase space* and the value $X(t)$ of a solution to Eq. 1 as the *state* of the system at time $t$. Our goal is to learn a $\gamma$-parameterized embedding map $z_i = \psi(F_i; \gamma)$ for each $F_i \in \mathcal{D}$ so that the $z_i \in \mathbb{R}^d$ capture the principal factors of variation in the underlying physical system given by Eq. 1 (Fig. 1). We reason that a good map, $\psi$, is one that produces embeddings, $z_i$, of testing data from which governing equations can be faithfully decoded. Much in the way word embeddings are learned by optimizing an embedding map on a self-supervised auxiliary task of context prediction (Mikolov et al., 2013), we train our embedding map on a self-supervised auxiliary task of governing equation prediction.

### 3.1 ARCHITECTURE

In practice, we evaluate each $F$ on a (potentially different) phase space lattice, each of whose $q$ dimensions has resolution $n$, so that $\mathcal{D}$ is a set of $n^q \times q$ arrays whose elements indicate the $q$-dimensional velocity of equation 1 at each discrete location. We set $n = 64$ but found no qualitative difference in performance when $n$ was varied between 32 and 128 (Fig. A.14). We instantiate the map, $\psi$, as three-layer convolutional architecture (with kernels having either 2 or 3 spatial dimensions depending on $q$) terminating in one fully-connected layer mapping convolutional features to the $d$-dimensional `phase2vec`.

Following (Brunton et al., 2016), embeddings are passed through a two-layer multi-layer perceptron, producing a vector of estimated coefficients, $\Xi_{recon} \in \mathbb{R}^{p \times q}$. These coefficients are used to form linear combinations with a set of basis functions $\Phi(X) = [\Phi_1(X), \ldots, \Phi_p(X)]$ for column vectors $\Phi_i$. The reconstructed equation is taken as

$$\dot{X}_{recon} = \Phi(X)\Xi_{recon} = F_{recon}(X, \Xi_{recon}) \tag{2}$$

where $\Phi(X) = \{x_j^{a_j}\}$ for $\sum_{j=1}^q a_j \leq c$ are all possible unit-coefficient monomials up to a given degree, $c$. We set $c = 3$ (cubic degree) so that for $q = 2$, $\Xi_{recon} \in \mathbb{R}^{10 \times 2}$ (Fig. 1). Other bases of larger polynomial degree or of non-polynomial functions, e.g. including $\sin(x)$, are also possible (Brunton et al., 2016). We chose $c = 3$ following earlier work (Iben & Wagner, 2021) which showed this value was sufficient for the types of dynamical systems considered here.

### 3.2 LOSS AND TRAINING

Following (Champion et al., 2019), we train our system with a loss that balances reconstruction fidelity with sparsity of the underlying equation. However, we also normalize our reconstruction loss in a manner that emphasizes special "skeletal" points of the dynamics. In particular, we take our reconstruction loss to be

$$\mathcal{L}_1(\dot{X}, \dot{X}_{recon}) = \frac{\|\dot{X} - \dot{X}_{recon}\|_2}{\|\dot{X}\|_2 + \epsilon}, \tag{3}$$

for a small corrective constant, $\epsilon > 0$. Here, $\mathcal{L}_1$ is especially large where $\dot{X}$ vanishes, lending special emphasis to fixed points and "slow" regions in the dynamics, which are often the loci of important dynamical phenomena (Tredicce & Lippi, 2004) (see Supp. A.1.1). These regions of low magnitude in an array only have meaning in the physical setting (as opposed to, for example, in image data), placing `phave2vec` under the "learning biases" rubric in physics-informed machine learning (Karniadakis et al. (2021), Box 2). We use a simple sparsity penalty $\mathcal{L}_2(\dot{X}) = \|\Xi_{recon}(\dot{X})\|_1$ and train $\psi(\dot{X}; \gamma)$ by approximating

$$\gamma^* = \arg\min_\gamma \mathbb{E}_{F \in \mathcal{D}} \left[ \mathcal{L}(\dot{X}, \dot{X}_{recon}) \right] \tag{4}$$

for $\mathcal{L} = \mathcal{L}_1 + \beta\mathcal{L}_2$ with sparsity regularizer $\beta$. Note that the loss measures the discrepancy between the true and reconstructed vector fields and not between the true and reconstructed parameters.

We investigated both $q = 2$ and $q = 3$ dimensional systems. The training set $\mathcal{D}$ in either case was a collection of $q$-dimensional polynomial ODEs generated from the dictionary $\Phi$ with coefficients 0 with probability .75 and otherwise sampled uniformly on $[-3, 3]$. We then tested generalization on two data sets, one measuring generalization to new parameter regimes for the training set and another measuring generalization to new functional forms not observed during training. For the former, we used a set of equations sampled from the same distribution generating $\mathcal{D}$ but having a disjoint set of coefficients, and, for the latter, we used a collection of "classical", real-world systems ranging from the FitzHugh-Nagumo neuron (FitzHugh, 1955) to the Lotka Volterra model (Freedman, 1980) to the Lorenz system (Lorenz, 1963). The functional form of equations from this latter set were systematically excluded from the training set. For example, if a testing system had the form $\dot{x_1} = a + bx_1^2$; $\dot{x_2} = cx_1x_2$, then this equation was never used to generate a training datum for any $a, b, c$ (see Sec. A.2).

## 4 RESULTS

**Generalization to held-out systems**   Embeddings were trained on an auxiliary task of equation reconstruction before being evaluated on a sequence of dynamics classification tasks. In order to ensure phase2vec could indeed learn this auxiliary task, we verified that we could reconstruct the equations of held-out dynamical systems compared to the simplest per-equation fitting baseline incorporating sparsity, LASSO (Tibshirani, 1996). The same sparsity regularizer of $\beta = 1 \times 10^{-3}$ was used in both models. We measured both the euclidean error from the true parameters and the fixed-point normalized reconstruction error (Eq. 3) on the vector fields.

| | Saddle-Node (1) | Pitchfork (1) | Transcritical (1) | Simple Oscillator (1) | Lotka-Volterra (1) |
|---|---|---|---|---|---|
| **Par: LASSO** | $2.15 \times 10^{-1}$ | $2.30 \times 10^{-1}$ | $2.33 \times 10^{-1}$ | $6.54 \times 10^{-1}$ | $7.00 \times 10^{-1}$ |
| **Par: Phase2Vec** | $1.60 \times 10^{-1}$ | $1.61 \times 10^{-1}$ | $1.78 \times 10^{-1}$ | $5.37 \times 10^{-1}$ | $4.51 \times 10^{-1}$ |
| **Recon: LASSO** | $4.16 \times 10^{-1}$ | $7.13 \times 10^{-3}$ | $4.12 \times 10^{-3}$ | $9.13 \times 10^{-3}$ | $1.99 \times 10^{0}$ |
| **Recon: Phase2Vec** | $1.75 \times 10^{-1}$ | $1.78 \times 10^{-1}$ | $1.71 \times 10^{-1}$ | $1.53 \times 10^{-1}$ | $1.26 \times 10^{0}$ |
| | **Homoclinic** (1) | **Van Der Pol** (1) | **Selkov** (2) | **FitzHugh-Nagumo** (4) | **Polynomial** (20) |
| **Par: LASSO** | $3.45 \times 10^{0}$ | $1.68 \times 10^{1}$ | $6.53 \times 10^{0}$ | $1.16 \times 10^{1}$ | $3.11 \times 10^{0}$ |
| **Par: Phase2Vec** | $2.93 \times 10^{0}$ | $1.15 \times 10^{-3}$ | $5.52 \times 10^{0}$ | $6.07 \times 10^{0}$ | $1.78 \times 10^{0}$ |
| **Recon: LASSO** | $2.32 \times 10^{-3}$ | $2.39 \times 10^{1}$ | $2.97 \times 10^{0}$ | $2.93 \times 10^{-1}$ | $5.26 \times 10^{-1}$ |
| **Recon: Phase2Vec** | $1.93 \times 10^{-1}$ | $3.26 \times 10^{-1}$ | $8.52 \times 10^{-1}$ | $4.62 \times 10^{-1}$ | $1.47 \times 10^{-1}$ |

Table 1: Euclidean parameter estimation error and reconstruction loss of phase2vec or LASSO on two-dimensional systems.

For two-dimensional systems (Table 1), our generalizable approach was found to outperform the per-equation LASSO baseline on average across the testing set of "classical" equations (see Sec. A.2; Parameter error: phase2vec, $2.70 \pm 0.29$ v.s. LASSO, $3.96 \pm 0.44$. Reconstruction error: phase2vec, $0.37 \pm 0.06$ v.s. LASSO, $0.6 \pm 0.12$; standard deviation bounds), al-

| | Saddle-Node 3d (1) | Lorenz System (3) |
|---|---|---|
| **Par: LASSO** | $7.16 \times 10^{-2}$ | $2.14 \times 10^{1}$ |
| **Par: Phase2Vec** | $5.65 \times 10^{-2}$ | $1.42 \times 10^{1}$ |
| **Recon: LASSO** | $1.83 \times 10^{-1}$ | $8.10 \times 10^{-1}$ |
| **Recon: Phase2Vec** | $2.28 \times 10^{-1}$ | $6.20 \times 10^{-1}$ |

Table 2: Euclidean parameter estimation error and reconstruction loss of phase2vec or LASSO for three-dimensional systems.

though performance per-class varied widely. These results were quantitatively maintained even when fixed-point normalization was turned off (Reconstruction error: phase2vec, $0.28 \pm 0.09$ v.s. LASSO, $0.89 \pm 0.15$). Further, in order to assess the ability of phase2vec to reconstruct systems which are not expressible in the output basis, we also included one system, the "simple oscillator", whose equation has no closed polynomial form (see Sec. A.2). Nevertheless, phase2vec was also found to fit this data comparably well to the LASSO baseline (Table 1, "Simple Oscillator (1)"). Again, none of the functional forms of the testing systems were observed during training, meaning that even out-of-distribution dynamics were encoded and decoded correctly.

To evaluate `phase2vec` embeddings in the three-dimensional case, we evaluated our architecture on equations having qualitatively similar behavior to the two-dimensional case (a three-dimensional extension of the saddle-node family; see Sec. A.2) as well as on equations giving rise to dynamics only possible for $q > 2$ (i.e. the chaotic behavior produced by the Lorenz system in three dimensions). As in the two-dimensional case, we found that three-dimensional reconstruction performance favored `phase2vec` (Parameter error: `phase2vec`, $4.78 \pm 0.27$ v.s. LASSO, $7.17 \pm 0.51$. Reconstruction error: `phase2vec`, $0.36 \pm 0.08$ v.s. LASSO, $0.39 \pm 0.15$; standard deviation bounds; see Table 2). An example reconstruction is given in Fig. A.15.

**Effects of normalization and sparsity**   We found that fixed-point normalization was crucial for accurate vector field reconstruction of testing data, especially for systems with complex dynamical features, such as cohabitating fixed points and cycles (Fig. 2a). For example, we compared the quality of reconstruction of the normalized and un-normalized models of a dynamical system exhibiting a homoclinic bifurcation in which a limit cycle collides with a saddle point as one of its coefficients is tuned from $\xi = -1.2$ to about $.8645$ (see Sec. A.2). We found that un-normalized reconstruction error was, naturally, lower for the model trained with the un-normalized loss. However, the ability of the normalized model to predict the ground truth system behavior near dynamical key points (such as fixed points) was much greater than the un-normalized model (Fig. 2a).   To

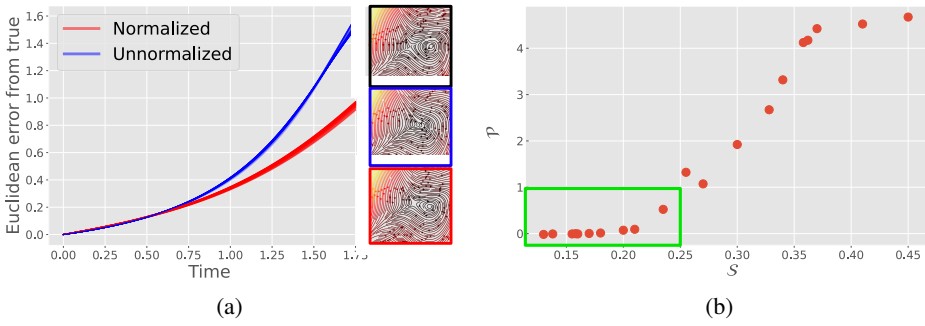

(a)                                                                  (b)

Figure 2: *Fixed-point normalization and sparsity. (a)* The euclidean distance between trajectories simulated by a `phase2vec` model trained with (red) or without (blue) physically-informed normalization, and the ground truth model, in the case of a system nearing a homoclinic bifurcation (Sec. A.2). Relative performance near the fixed point is evident from stream plot reconstructions (black inset: ground true; blue inset: unnormalized; red inset: normalized.) *(b)* Parameter prediction error, $\mathcal{P}$, as a function of the proportion of ground truth (noisy) parameters sparsified, $\mathcal{S}$, over different settings of the sparsity regularizer, $\beta$. Low reconstruction error is obtained despite sparsification up to $25\%$ (green box).

observe the necessity of the normalization more closely, we generated 20 reconstructions for each of the normalized and un-normalized model corresponding to 20 evenly spaced parameter values in the interval $\xi \in [-1.2, -.8645]$. We then simulated 1000 trajectories from all of these systems with initial conditions sampled from a circle of radius $.1$ surrounding the fixed point at the origin. As the trajectories advanced in time, we measured the euclidean deviation of simulated vs true trajectories, averaged across all initial conditions (Fig. 2a). We found that trajectories starting near the fixed point diverged much more rapidly from their true course when simulated based on the un-normalized model compared to those based on the normalized model. This is especially noticeable when the reconstructions are viewed as streamline plots (Fig. 2a: ground truth, black border inset; blue border inset, un-normalized reconstruction, and red border inset, normalized-reconstruction). In this sense, the normalized loss is critical for inferring meaningful dynamical features while ignoring irrelevant details which influence the euclidean loss (details in Sec. A.1.1).

Furthermore, we expect meaningful and useful representations of dynamical systems to correspond to the simplest system that generates the underlying input dynamics, i.e. the system with the sparsest parameter values. Therefore, we next investigated the role of sparsity in the auxiliary equation prediction task by measuring parameter reconstruction error on a new version of the classical system testing set (i.e. the systems of Table 1) whose underlying parameters, $\Xi_{pert}$, were perturbed by zero-mean gaussian noise with standard deviation $\sigma = .1$. We varied $\beta$ from $1 \times 10^{-3}$ to $1 \times 10^{-1}$ in

20 steps and for each $\beta$ measured the average euclidean distance between the inferred parameters and the true (unperturbed) parameters, $\mathcal{P}$, versus the average proportion of the perturbed parameters which were sparsified, $\mathcal{S}$:

$$\mathcal{P} = \mathbb{E}_{\mu}\left[\|\Xi_{recon} - \Xi\|_2\right], \quad \mathcal{S} = \mathbb{E}_{\mu}\left[\frac{\|\Xi_{recon}\|_1}{\|\Xi_{pert}\|_1}\right]. \tag{5}$$

for $\Xi_{pert} \sim \mu$. We found that the perturbed parameters could be substantially sparsified (approximately over the range $\beta \in [1 \times 10^{-3}, 1 \times 10^{-2}]$) without incurring a large reconstruction error (Fig. 2b, green box). This provides evidence that the model has learned to distinguish between parameters that contribute to dynamical structures of the systems and irrelevant nuisance parameters.

**Dynamically-informed reconstruction of noisy test data**  The applicability of our model to real data depends on our ability to produce meaningful embeddings from noisy data, as vector fields are typically acquired by binning derivatives from time series datasets which can be sparse or corrupted. We do not propose `phase2vec` as a denoiser *per se*, rather intending to show that our learned representations are robust to noise compared to a per-equation baseline. To evaluate this ability, we measured reconstruction performance on the "classical" system testing data subjected to four types of noise: (1) independent, zero-mean gaussian noise added to input vector fields, (2) random zero masking, (3) random sparsification arising from creating vector fields from binned trajectory data and (4) gaussian noise added to the true parameter vector (for details, see Sec. A.2). The magnitude of each type of noise was systematically varied as we probed the comparative ability of `phase2vec` vs LASSO to reconstruct the uncorrupted data.

We found that `phase2vec` reconstructions degraded gracefully with noise compared to LASSO over a wide range of noise parameter values (Fig. 3). For examples of `phase2vec` reconstructions, see Sec. A.4. Crucially, as these were testing data, reconstruction is only possible if the embedding map, $\psi$, has acquired a robust and general notion of how the geometry of vectors in phase space relates to governing equations.

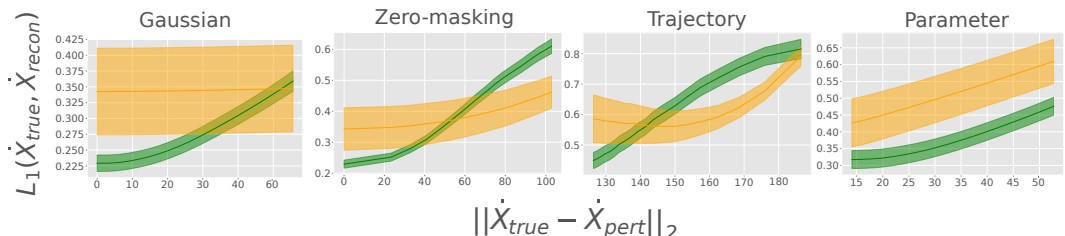

Figure 3: *Dynamically-informed denoising.* The accuracy of `phase2vec` (green) in comparison to LASSO (yellow) in reconstructing noisy testing data in four cases. From left to right: gaussian noise applied to the input vector field, random zero masking applied to the input vector field, trajectory-generated noise, and gaussian noise applied to the true parameter vector. For low values of noise `phase2vec` was always the more robust method. For parameter noise, this trend continued for the full range of noise examined.

**Decoding the physics of embedded data**  Having ensured that governing equations for testing data could be decoded from `phase2vec` embeddings with performance comparable to a per-equation fitting method including under noisy conditions, we next sought to validate the quality of these embeddings on a battery of physics classification tasks. We compared `phase2vec` to three other representations: (1) the parameter vector of the underlying equation, (2) the first $d$ PCA eigenvalues of the raw vector fields where $d$ is set to be the dimension of the embedding space, and (3) time series features extracted by TapNet (Zhang et al., 2020), a state-of-the-art time series classifier. For all representations, except for TapNet (which has a built-in MLP classifier), we trained a logistic regressor with a cross-validated $\ell_2$ penalty to predict the true classes of the underlying dynamics. We ran three experiments with classes delineated according to either global or local dynamical properties. For the global case, we classified: (1) whether the dynamics respect conservation of energy or not (Conservativty; two classes), and (2) whether the dynamics

represent an incompressible flow or not (Incompressibility; two classes)[2] For the local case, we classified linear systems according to the five possible types of fixed points they can exhibit (Linear stability; five classes: stable node, unstable node, stable spiral, unstable spiral or saddle point (Strogatz et al., 1994)). TapNet classifications came from voting over classifications made from 10 length-64 time-series. All systems were two-dimensional. For training details, see Sec. A.3.

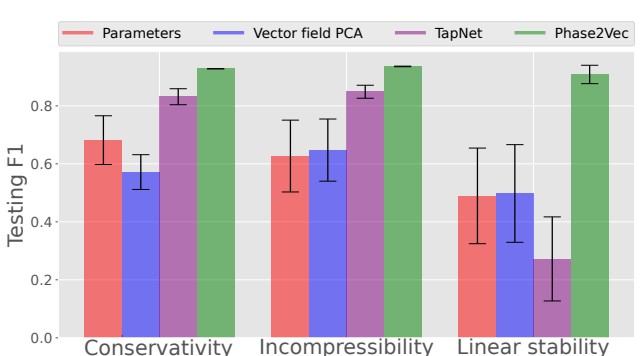

On all classification tasks, we found that phase2vec embeddings outperformed the competing representations (Fig. 4), achieving an F1 score of over .9[3]. The largest comparative advantage for phase2vec was achieved on the five-class linear stability task, where the otherwise high-performing TapNet achieved an F1 score of under .3. The relative advantage of phase2vec over Tap-Net specifically in the cases of conservation of energy and incompressibility demonstrates one of the important advantages of a vector-field-based approach. Namely, these are global system properties which are naturally captured by the extensive coverage of phase space as well as the aggregation of phase information which are entailed by convolutional features.

Figure 4: *Classification performance of phase2vec vs. alternative representations.* In physics classification, phase2vec (green) outperformed competing representations: (1) the raw 20-dimensional parameter vector of the true equation (red), (2) the first $d = 100$ PCA eigenvalues of the input vector field (blue), and (3) features extracted using the attentional time series model TapNet (purple). Bars depict F1 score and errors are standard deviation across classes. Details in main text and Sec. A.3).

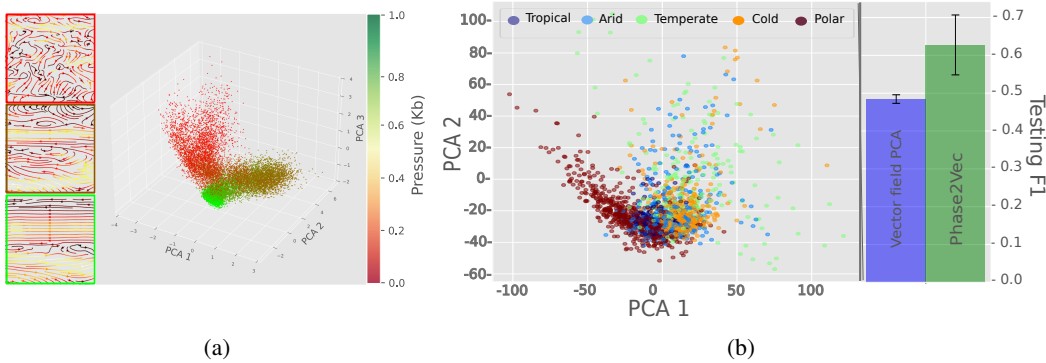

|   |   |
|---|---|
| (a) | (b) |

Figure 5: *Climate embeddings. (a) Emergent clusters in wind data.* Each point is a phase2vec embedding of a vector field measuring wind velocity in a fixed spatial grid covering the eastern hemisphere. Different points correspond to different times and pressures. Points are colored by pressure from .01 to 1.0 Kb. Three clusters emerge, corresponding to the appearance of circumglobal streamlines (insets) as a function of pressure. *(b) Temperature gradient embeddings and Köppen label prediction.* Temperature gradient embeddings colored according to a coarse Köppen label, indicating the general climatic zone which predominates in the crop. Labels were better predicted using phase2vec than using raw vector fields matched for dimensionality by PCA.

---

[2]These are two important dynamical classes since Helmholtz's Theorem holds that, under mild assumptions, every vector field can be decomposed as the sum of a conservative and incompressible component.

[3]We also confirmed that we could decode the identities of the "classical" systems used above as test reconstruction data (F1 score, .814).

**Embeddings of climate data**    To investigate whether our embeddings could identify meaningful structure in real data, we computed representations of two types of climate data. For a first, qualitative, demonstration, we computed embeddings for global wind vectors taken from (Blumenthal et al., 2005). Measurements were indexed by month and by year, from 1960 to 2022, as well as by pressure (Kb). We used wind vector measurements from a fixed spatial grid comprising roughly the eastern hemisphere which exhibits distinct visual patterns in the form of horizontal streamlines emerging at low pressures (Fig. 5a; brown inset for .07 Kb and green inset for .01 Kb). We found that `phase2vec` embeddings could easily identify these pressure-dependent bands in the form of the red, brown, and green clusters (Fig. 5a).

For a second, quantitative experiment, we computed embeddings for world average temperature data taken from (Fick & Hijmans, 2017) which were accompanied by so-called Köppen labels which indicate to which of five coarse temperature zones a given spatial location belongs (Köppen, 1884; Beck et al., 2018). The ability to make quantitative predictions about these data is especially important since they have been recently shown to produce accurate forecasts of changing climate zones as a result of global warming (Beck et al., 2018).

To demonstrate the ability of `phase2vec` to classify data taken from a real physical system, we generated embeddings of random spatial crops (sized $153° \times 153°$ latitude by longitude) of temperature gradient fields and used them to predict each crop's expected Köppen label. As these fields were indexed by month and labels were not, we used month-averaged embeddings as our representation of a given location. We found that `phase2vec` embeddings of these data (first two PCs, Fig. 5b, left) were substantially more predictive of the expected Köppen label than the raw vector field (Fig. 5b right, VF F1 testing score: $.484 \pm .011$ vs `phase2vec` F1 testing score: $.627 \pm .079$), indicating that `phase2vec` had extracted dynamical structure relevant to climate annotations.

## 5    CONCLUSION

`phase2vec` is a physics-informed convolutional network for high-quality, low-dimensional representations of dynamical systems. Learned features are trained on an auxiliary task of generalized equation prediction and are demonstrated to be robust to noise. Importantly, `phase2vec` can generalize to held-out systems and learn the underlying semantics of dynamical systems, which can be used to decode general physical characteristics from data with greater fidelity than competing methods.

A clear area for future work is the extension of our framework to high-dimensional systems. This could be accomplished in a straightforward way with the use of higher-dimensional convolutions, which have been used up to several tens of dimensions (Choy et al., 2020). For higher dimensional real-world systems, like neural circuits or chemical signaling pathways, potential approaches could use architectures taking advantage of sparse measurements in phase space, such as graph neural networks.

In its current form, `phase2vec` makes minimal assumptions about input data, namely, that dynamics are qualitatively invariant to translations of the input coordinate system, that "slow" dynamics are important, and that dynamics generally change stably with input parameters. The first two of these assumptions are encoded by our use of a convolutional architecture and fixed-point normalization, respectively. The stability assumption is encoded by the fact that the embedding map is continuous and validated by showing that embeddings are robust to noise. Future work could introduce new assumptions, for instance by enforcing additional types of symmetry (e.g. rotational symmetry). Existing assumptions could also be adapted, for instance, by making embeddings sensitive to qualitative changes in dynamics (i.e. bifurcations, see (Kuznetsov, 1998)) which result from otherwise small changes in parameters. The choice of dictionary functions is also an area for future exploration, and we are intrigued by the use of other bases, like Legendre polynomials or biologically-meaningful components, like Hill functions (Frank, 2013; Ingalls, 2013).

The ability of `phase2vec` to encode meaningful dynamical features from data makes it a useful and promising model for the recovery of minimal physical systems and the design of new dynamical systems across biological and engineering applications. Chief among these applications are the design and real-time control of dynamical systems across the sciences.

ACKNOWLEDGMENTS

We thank Bianca Dumitrascu and our group members for discussions and feedback. This work was supported by the Zuckerman Postdoctoral Program (M.R.), the Israeli Council for Higher Education Ph.D. fellowship (N.M. and Z.P.), the Center for Interdisciplinary Data Science Research at the Hebrew University of Jerusalem (N.M.), Clore Scholarship for Ph.D. students (Z.P.), Azrieli Foundation Early Career Faculty Fellowship, and the European Union (ERC, DecodeSC, 101040660) (M.N.). Views and opinions expressed are however those of the author(s) only and do not necessarily reflect those of the European Union or the European Research Council. Neither the European Union nor the granting authority can be held responsible for them.

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

# A  APPENDIX

All experiments were carried out using `pytorch` v. 1.12 using an NVIDIA RTX 3060 GPU.

## A.1  ARCHITECTURE AND TRAINING

Here, we describe the embedding architecture of `phase2vec` specifically for the two-dimensional case ($q = 2$). The three-dimensional case is identical, except that the number of spatial dimensions of kernels is one larger ($q = 3$) and the embedding dimension, $d$, was larger. The convolutional part of the `phase2vec` embedding pipeline consisted of three convolutional blocks, each with kernel size $3 \times 3$, stride $2 \times 2$ and 128 channels. This resulted in a sequence of convolutional blocks of sizes $(128 \times 31 \times 31)$, $(128 \times 15 \times 15)$, $(128 \times 7 \times 7)$. Convolutional blocks were interleaved by ReLU non-linearities and batch norm layers. The final convolutional activations were mapped by a single linear layer to a $d = 100$-dimensional embedding space ($d = 256$ for three-dimensional case).

To compute predicted coefficients, these embeddings were mapped by a 2-layer MLP whose hidden layers had 128 units to 20-dimensional vectors representing the 10 coefficients possible for each of the 2 dimensions in the ODE. For the three-dimensional case, embeddings were mapped to 60-dimensional vectors representing 20 coefficients for each of the 3 dimensions. MLP layers were alternated with batch norm layers as well as dropout layers whose rate we set to $p = .1$ during training.

The model was trained with an ADAM optimizer using a learning rate of $1 \times 10^{-4}$ over 200 iterations.

### A.1.1  PHYSICALLY-INFORMED LOSS NORMALIZATION

We provide a visualization of the physically-informed normalized loss in comparison to the unnormalized loss showcasing its importance in capturing meaningful dynamics. Considering the saddle-node bifurcation system Fig. A.6a we observe that the normalized loss landscape across phase space captures the properties of the system, visually represented by the fixed points Fig. A.6b. In contrast, these can not be identified in the unnormalized representation Fig. A.6c.

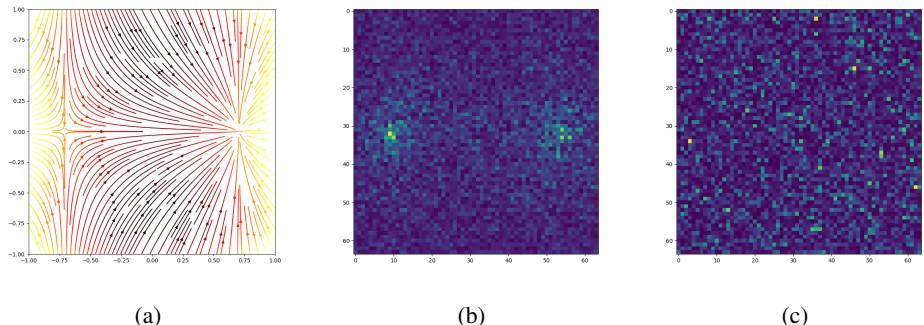

(a)           (b)           (c)

Figure A.6: *Fixed-point normalization.* *(a)* Saddle-node vector field used for evaluation of the loss landscape *(b)-(c)*. Gaussian noise was added to the true vector field and losses were measured at each spatial location. *(b)* shows the physically-informed normalization and *(c)* the unnormalized loss.

## A.2  DATA, TRAINING AND EVALUATION

### A.2.1  SYNTHETIC DATA

The training set $\mathcal{D}$ was a collection of polynomial ODEs generated from the dictionary $\Phi$ with coefficients that were 0 with probability .75 and were otherwise sampled uniformly on $[-3, 3]$ and consisted of 10k examples. For evaluation in all settings, a test set of 1000 samples was used.

Network hyperparameters were selected to minimize validation loss on two data sets:

1. *Polynomial equations*: For generalization to new parameter regimes. A set of equations sampled from the same distribution generating $\mathcal{D}$ but having a disjoint set of coefficients.

2. *Real-world systems*: For generalization to new functional forms not observed during training. The real-world systems, their parameter ranges and labels are given in Table A.3.

All equations were re-centered to be on the square $[-1, 1]^2$ on the phase plane and discrete vector fields were computed by measuring the continuous system on an evenly-spaced grid of $64 \times 64$ points. We only investigated equation reconstruction for two 3-d systems as a proof of concept and did not classify them. They therefore do not have labels.

| Name | Equation | Parameter ranges | Class labels |
|---|---|---|---|
| Saddle-node | $\dot{x_1} = a - x^2; \dot{x_2} = -1$ | $a \in [-1, 1]$ | 0 if $a < 0$; 1 if $a \geq 0$ |
| Pitchfork | $\dot{x_1} = ax_1 - x_1^3; \dot{x_2} = -1$ | $a \in [-1, 1]$ | 2 if $a < 0$; 3 if $a \geq 0$ |
| Transcritical | $\dot{x_1} = ax_1 - x_1^2; \dot{x_2} = -1$ | $a \in [-1, 1]$ | 4 if $a < 0$; 5 if $a \geq 0$ |
| Simple Oscillator | $\dot{r} = r(a - r^2); \dot{\theta} = -1$ (polar) | $a \in [-1, 1]$ | 6 if $a < 0$; 7 if $a \geq 0$ |
| Lotka-Volterra | $\dot{x_1} = x_1(1 - x_2); \dot{x_2} = ax_2(x_1 - 1)$ | $a \in [-1, 1]$ | 8 for all $a$ |
| Homoclinic | $\dot{x_1} = x_2; \dot{x_2} = ax_2 + x_1 - x_1^2 + x_1 x_2$ | $a \in [-1.2, -.7]$ | 9 if $a < -.8645$; 10 if $a \geq -.8645$ |
| Van Der Pol | $\dot{x_1} = x_2; \dot{x_2} = a(1 - x_1^2)x_2 - x_1$ | $a \in [.1, 4]$ | 11 for all $a$ |
| Selkov | $\dot{x_1} = x_1 + ax_2 + x_1^2 x_2; \dot{x_2} = b - ax_2 - x_1^2 x_2$ | $a \in [.05, .15], b \in [.2, 1.0]$ | 12 if $a < .3$; 13 if $a \geq .3$ |
| FitzHugh-Nagumo | $\dot{x_1} = x_1 - \frac{x_1^3}{3} - x_2 + a; \dot{x_2} = \frac{1}{b}(x_1 + c - dx_2)$ | $a \in [.1, .5]; b \in [10, 15]; c \in [.6, .7]; d \in [.7, .8]$ | 14 if $a < .35$; 15 if $a \geq .35$ |
| Saddle-node (3-d) | $\dot{x_1} = a - x^2; \dot{x_2} = -1; \dot{x_3} = -1$ | $a \in [-1, 1]$ | N/A |
| Lorenz (3-d) | $\dot{x_1} = a(x_2 - x_1); \dot{x_2} = x_1(b - x_3) - x_2; \dot{x_3} = x_1 x_2 - cx_3$ | $a \in [9, 11]; b \in [14, 28]; c \in [2, 4]$ | N/A |

Table A.3: Real-world systems, their equations and class boundaries. Systems having two classes exhibit a bifurcation (i.e. the emergence of a new topological structure in the dynamics: a fixed point, limit cycle, etc.)

### A.2.2 REAL DATA

**Global wind vectors**   Data was downloaded from IRI/LDEO Climate Data Library (`https://iridl.ldeo.columbia.edu/`). Measurements were indexed by month and by year, from 1960 to 2022, as well as by pressure (Kb). We used wind vector measurements from a fixed spatial grid comprising roughly the entire eastern hemisphere. A set of 2000 samples was used for testing.

**World climate data**   We took climate data from WorldClim V2 (`http://www.worldclim.org`). We used monthly temperature averages, taken over a temporal span of 1970–2000 at a spatial resolution of $0.083°$. Corresponding Köppen labels were downloaded from GloH2O, Köppen-Geiger (`http://www.gloh2o.org/koppen/`). We used a coarse-grained level of the labels amounting to five temperature zones (tropical, arid, temperate, cold and polar). A set of $2000 \times 12$ (per month) samples was used for testing.

### A.3 PHYSICS CLASSIFICATION

All data sets had 1000 samples with equal numbers of class exemplars. Embeddings were acquired for all data using the model trained on polynomials systems as described above. We considered three problems:

**Conservativity**   A conservative system represents a physical system which respects conservation of energy. Every conservative system can be represented as the gradient of a scalar field. To generate conservative systems, we first generated scalar fields by sampling the coefficients of a single polynomial equation with basis functions in $\Phi$ according to the same distribution as used to generate training coefficients. We then manually calculated gradients of these scalar fields. The opposing class were 500 samples from the training set which we ensured were not conservative. We did so by calculating the curl of each counter datum and checking it was nonzero (on a simple connected phase space a system is conservative if and only if it is irrotational, i.e., has zero curl.

**Incompressibility**   A system is incompressible if it represents the flow of a fluid which cannot be "compressed" into or out of phase space. In other words, the divergence of the field is zero. To generate divergence-free fields, we first identified the phase plane with $\mathbb{C}$ so that each point $(x_1, x_2)$

was identified with $z = x_1 + ix_2$. We again created a polynomial (automatically holomorphic) using the same distribution on parameters and having the form

$$f(z) = f(x_1 + ix_2) = a + ib. \tag{6}$$

We then defined the vector field $\frac{df}{dz} = v - iw$ where

$$v = \frac{\partial a}{\partial x_1} \tag{7}$$

and

$$w = \frac{\partial a}{\partial x_2}. \tag{8}$$

The component $b$ automatically satisfies the equivalence of mixed partials,

$$\frac{\partial^2 b}{\partial x_1 \partial x_2} = \frac{\partial^2 b}{\partial x_2 \partial x_1}, \tag{9}$$

which gives $\nabla \dot{v} = 0$ and so $v$ was taken as our divergence-free, incompressible system.

**Linear stability**  A linear planar system $\dot{x} = Ax$ exhibits a stable node, unstable node, stable spiral, unstable spiral or saddle point as a function of the trace and determinant of $A$. We randomly sampled $A$ until 200 exemplars of each stability class were acquired.

For each task we computed the relevant embeddings and then trained a logistic regressor on $80\%$ of the data. The remaining $20\%$ was held out for testing, using a stratified train-test split. An $\ell_2$ penalty was cross-validated using leave-one-out cross-validation on validation data split from the training set. We searched for an optimal regularizer over 11 values spread logarithmically between $1 \times 10^{-5}$ and $1 \times 10^5$. For multi-class settings, we used a one-versus-rest scheme.

### A.4 VECTOR FIELD RECONSTRUCTIONS

We provide `phase2vec` reconstructions over test data. Starting with a demonstration of the different classes, (1) Conservative (Fig. A.7) (2) Incompressible (Fig. A.8) and (3) Linear (Fig. A.9).

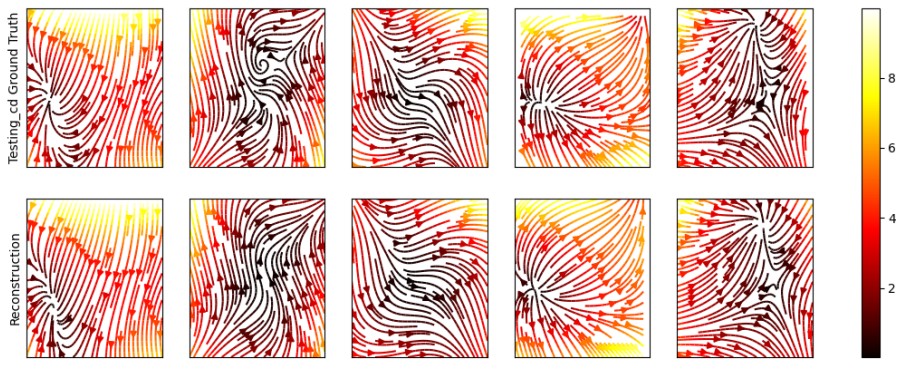

Figure A.7: Conservative dynamics, ground truth vs. reconstruction

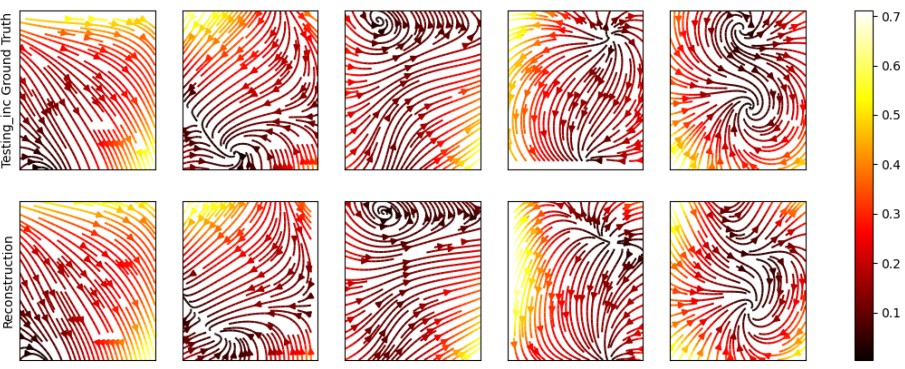

Figure A.8: Incompressible dynamics, ground truth vs. reconstruction

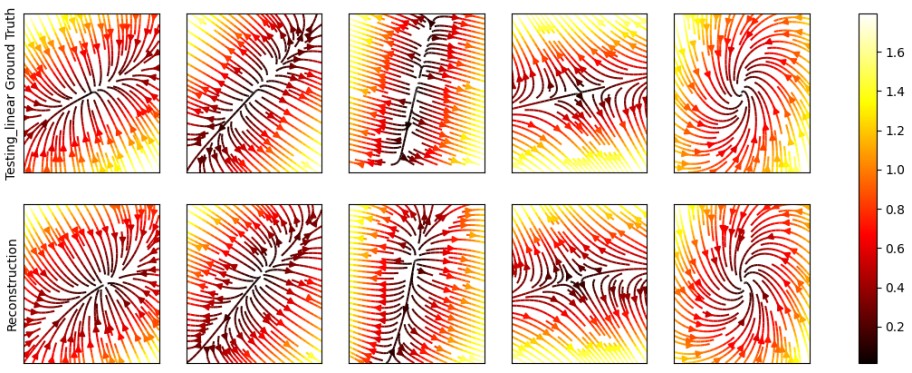

Figure A.9: Linear stability dynamics, ground truth vs. reconstruction

To assess the stability of phase2vec to noise, we perturbed testing data with four types of noise and compared the ability of our method to reconstruct the unperturbed data to that of a LASSO baseline. These noise types were:

- Gaussian: Zero-mean, independent gaussian noise was added to the vector fields. The standard deviation of the noise was set to $\sigma * \sigma_{true}$, where $\sigma_{true}$ was the true standard deviation of the data to be perturbed. This was done to scale the noise to each data class. We varied $\sigma$ from 0 to .3 in 20 steps.

- Zero-masking: A proportion of each vector field was randomly zero-masked. The proportion was varied between 0 and .3 in 20 steps.

- Trajectory: Data was generated by simulating a certain number of trajectories which were run for 100 steps with a step size of .01 and then calculating velocities by binning and averaging. The number of trajectories was used to control the amount of noise. We used between 10 and 2000 random initial conditions in 20 steps. Empty bins were filled with zeros.

- Parameter: We added zero-mean gaussian noise to the true parameter vector of the data, varying the standard deviation between 0 and .3 in 20 steps.

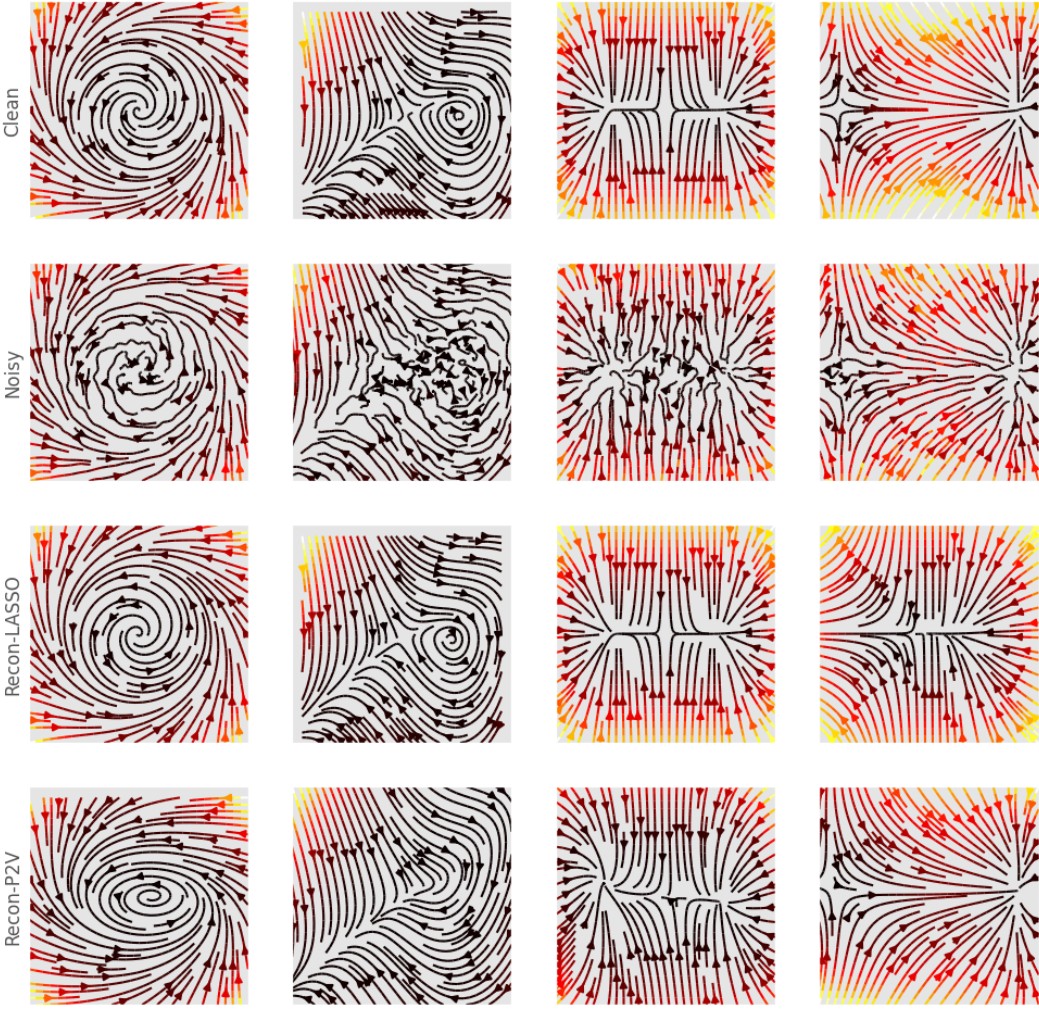

Figure A.10: Added gaussian noise of order of .3 times the ground-truth standard deviation of each vector field. Rows, top to bottom, depict the ground truth dynamics, noisy, LASSO reconstruction and *phase2vec* reconstruction.

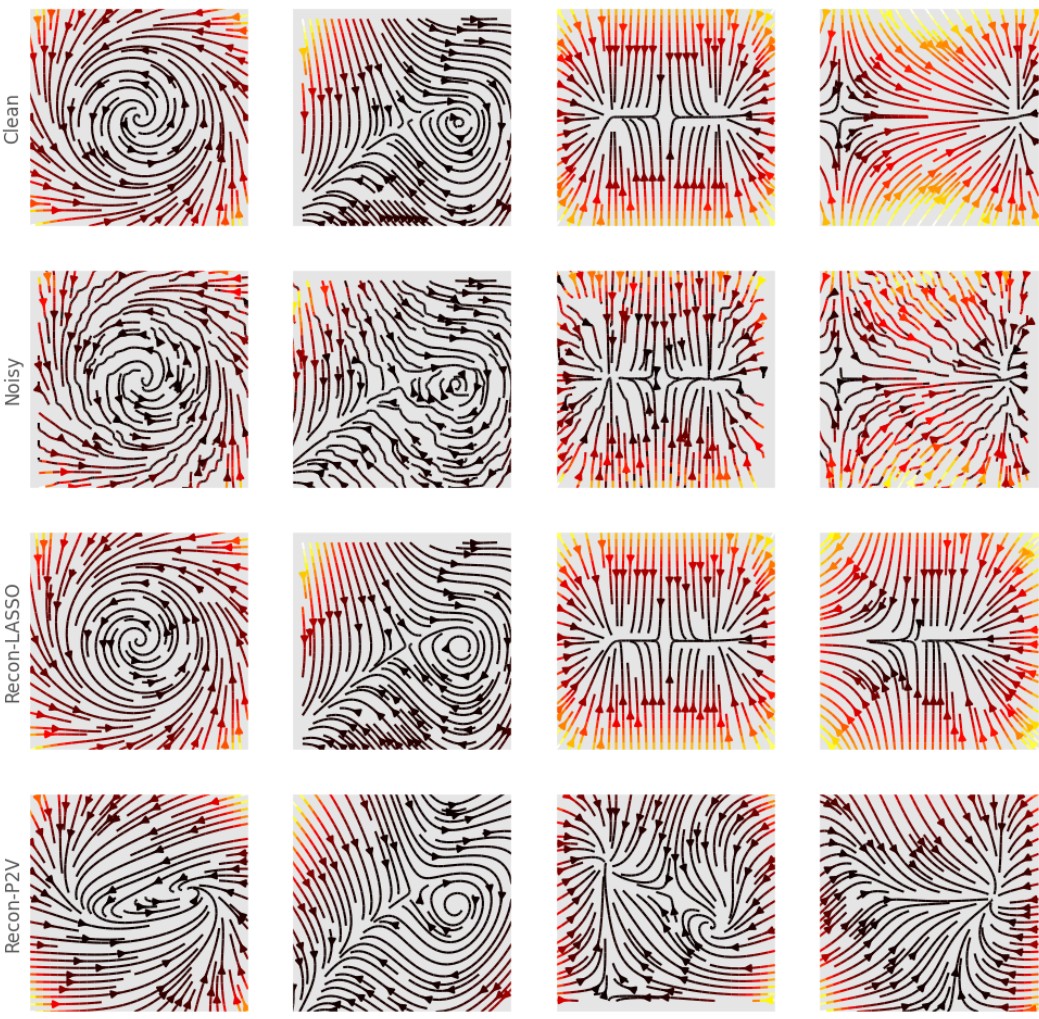

Figure A.11: Random zeroing out of 30% of the vectors. Rows, top to bottom, depict the ground truth dynamics, noisy, LASSO reconstruction and *phase2vec* reconstruction.

Last, we consider the reconstruction for different resolutions of the data, taking only $n = 32, 64, 128$ lattice points. Of note, the losses averaged over the simple oscillator test case were identical, but it took longer to train each resolution: 50, 100 and 150 iterations respectively.

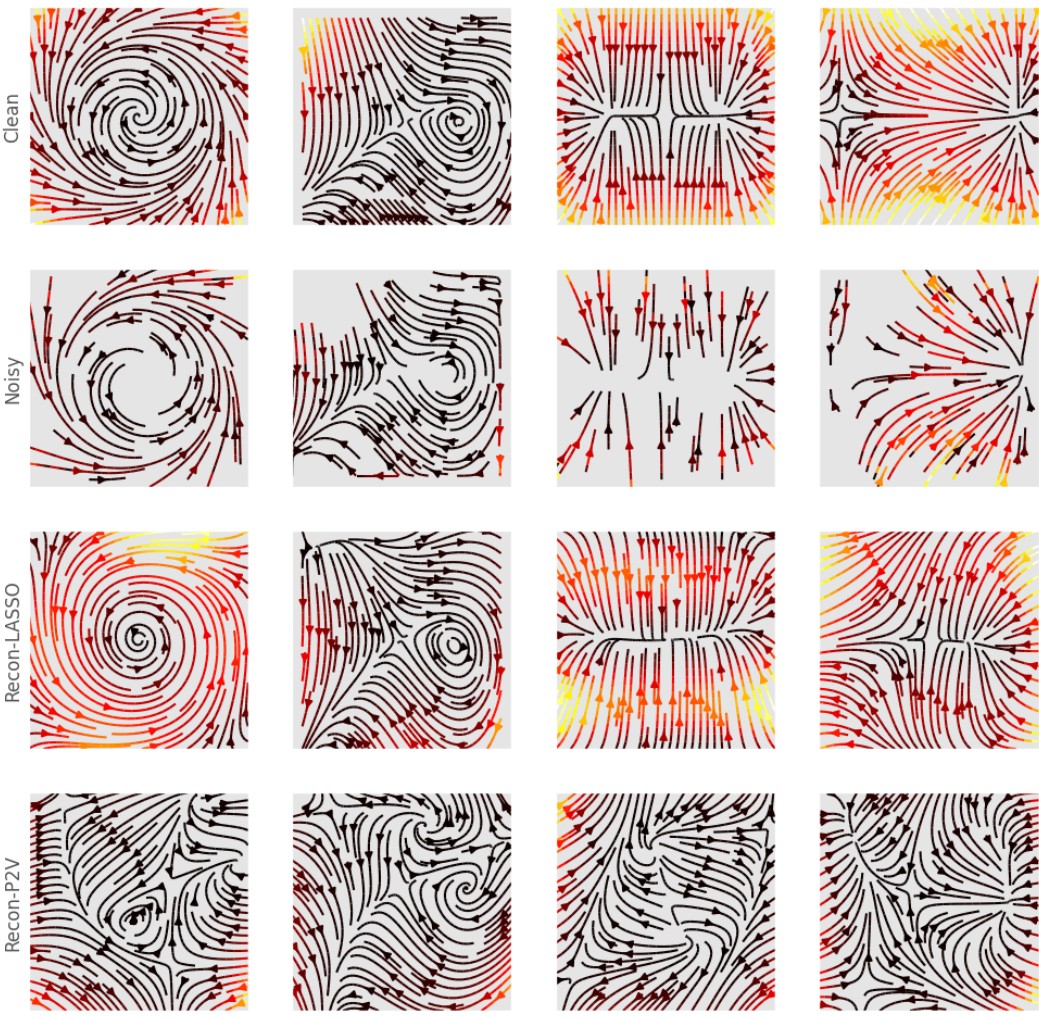

Figure A.12: Reconstruction based on limited trajectory set (100 initial conditions). Rows, top to bottom, depict the ground truth dynamics, noisy, LASSO reconstruction and *phase2vec* reconstruction.

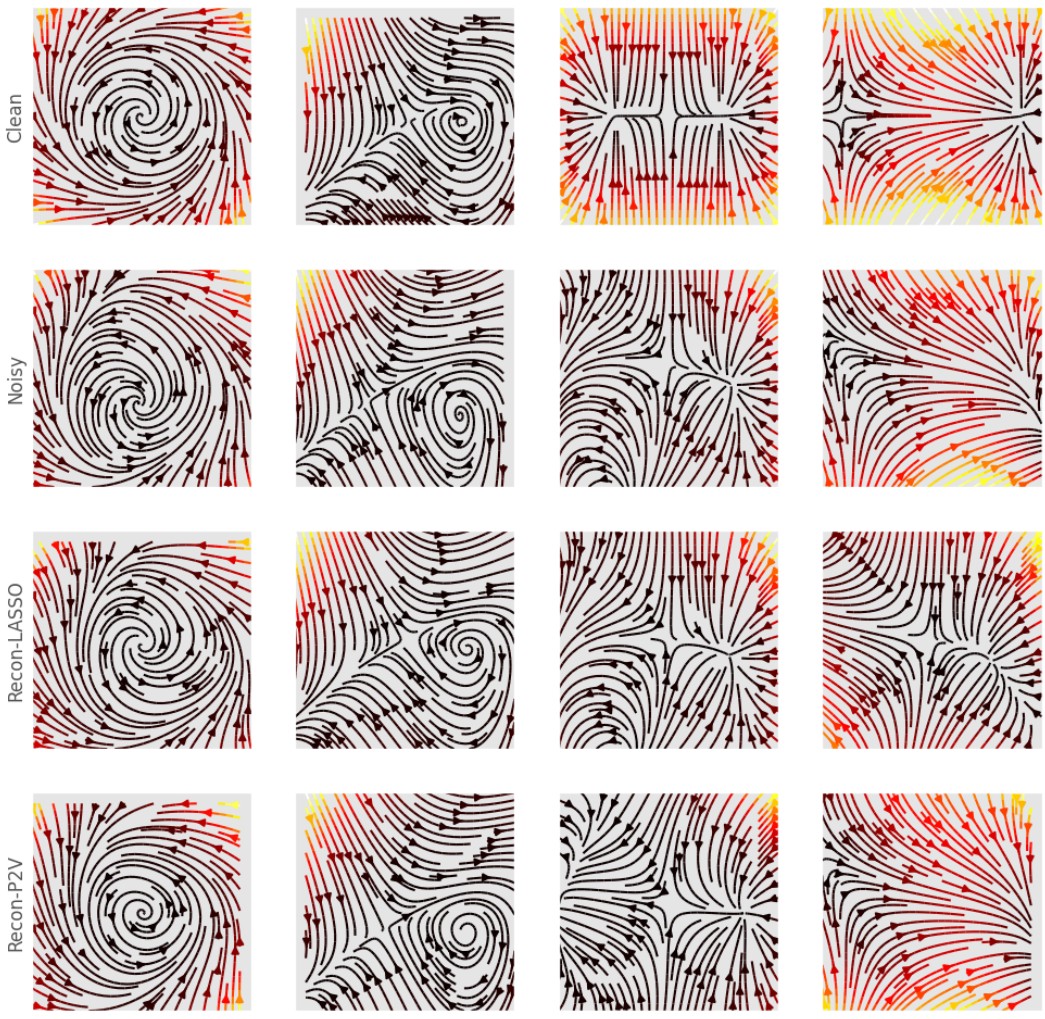

Figure A.13: Added gaussian noise of order of .3 times the standard deviation of the true parameter vector to the parameters vector. Rows, top to bottom, depict the ground truth dynamics, noisy, LASSO reconstruction and *phase2vec* reconstruction.

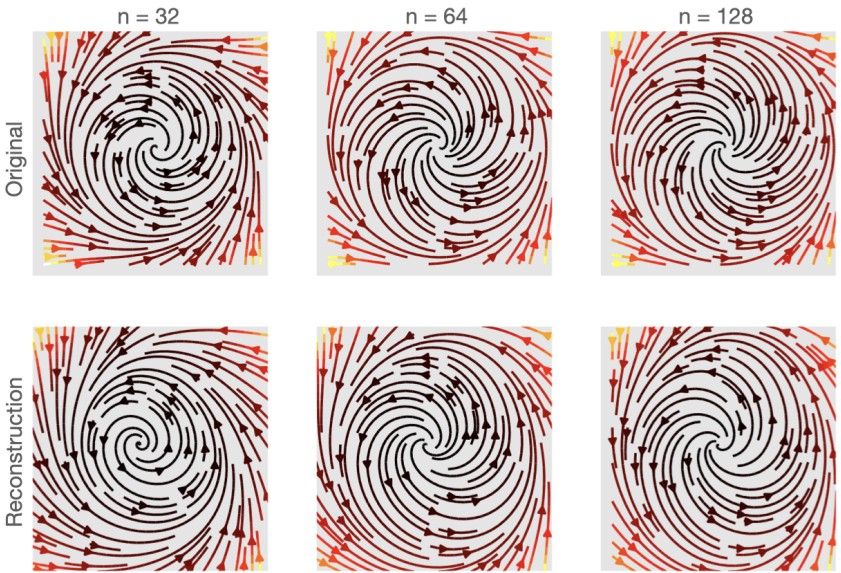

Figure A.14: Reconstruction based on different resolutions of the data. Columns to correspond to the number of lattice points taken, $n = 32$ (left) $n = 64$ (center) $n = 128$ (right). The top row presents the ground truth dynamics and the bottom row the reconstruction

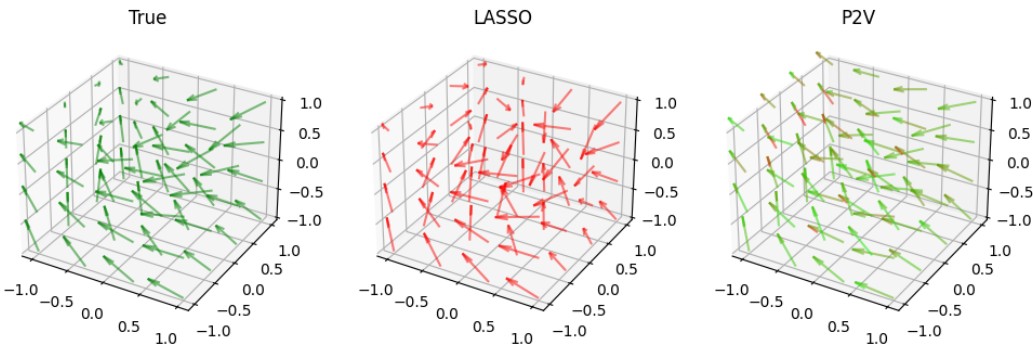

Figure A.15: *Example reconstructions.* Three vector fields from a 3-d saddle-node system are depicted. Arrows represent velocity at a given location and their color represents deviation from the true velocity, with red being the highest error and green being the lowest error. *(Left)* Ground truth (i.e. zero error); *(Center)* LASSO reconstruction having high error; *(Right)* phase2vec reconstruction with relatively low error.

