# OpenReview forum: "Phase2vec: dynamical systems embedding with a physics-informed convolutional network"
_ICLR.cc/2023/Conference — ICLR 2023 notable top 25%_

### Official Review · Reviewer_STRm · 2022-10-24

**Confidence:** 4
**Correctness:** 4
**Technical Novelty And Significance:** 3
**Empirical Novelty And Significance:** 3
**Recommendation:** 5

**Clarity, Quality, Novelty And Reproducibility:**

The paper is well written (with a minor exception for the last section), seems interesting and novel yet not very efficient, and limited to rather simple dynamical systems. However, reproducibility is not straightforward. Extensive description of the model should be added in the appendix. Code release would have been appreciated too.

**Strength And Weaknesses:**

The paper introduces an interesting method for the study of dynamical systems. One of the key points of the approach is the ability to generalize to different dynamical systems without requiring re-training, which is not the case for existing methods. The network architecture seems reasonable, as does the design of the criterion.

However, I believe the following points are debatable:

- Embeddings are learned using a phase field reconstruction task, through a sparse regression to the coefficients of the dynamic equation. This has several limitations in my opinion:
    - This requires the supervision of a relatively complete dynamics map, which is not easy to access in many situations. However, the authors show that their method is sufficiently robust to withstand failures and errors in the phase field.
    - Predictions are limited to the structure of the function dictionary used $\Phi_\alpha$. Since this contains only polynomial structures, a large number of dynamical systems become out of reach. Can the authors discuss this limitation in a more general framework?

- The use of convolution greatly limits the number of systems that can be studied, since the dynamics must be of dimension 2. Convolution can possibly be extended to higher dimensions, but this becomes non-trivial. This is for me one of the most negative points of the approach.

- The experiments are for me rather unconvincing, in particular the reconstruction of the dynamics. Of the 10 systems studied, LASSO greatly outperforms Phase2Vec on 4 systems. When Phase2vec dominates, the gap with LASSO is smaller. However, I acknowledge that Phase2Vec do not requires training to infer on unseen dynamics, while it is not the case for LASSO. The denoising and masking experiments are very interesting, but quite unrealistic for me. Would it be more relevant to generate the phase field from a set of trajectories whose initial conditions will be chosen at random? This will make the phase field more representative of what can be obtained in practice.

- I would also be curious to see how the network behaves beyond the training set, i.e. outside the spatial limits of the phase plane used to estimate the embedding.

MINOR
- I found experiments on real world data a bit hard to read.
- 3.1: "at a discrete LOCATION"

**Summary Of The Paper:**

This paper introduces a method similar to word2vec but applied to dynamic systems. The approach considers the phase space as an image, from which we can extract a latent representation summarizing the characteristics of the dynamics.

**Summary Of The Review:**

This paper introduces an interesting idea, but too limited on the number of possible applications. It is, however, an important line of research, which deserves to be taken into account.

---

> ### Author Response · Authors · 2022-11-18
> **Response to reviewer STRm**
>
> We thank the reviewer for these remarks which have resulted in substantial changes to the manuscript. Most notably, we were encouraged by the reviewer’s remarks to improve model performance by training with a lower learning rate (1e-4 vs 1e-3) over a longer period (200 vs 100 iterations) and to evaluate the model on a larger training set (10000 vs 1000 examples of “classical” systems). We found that these improved embeddings allowed us to outperform the LASSO baseline and the manuscript has been adjusted accordingly (Sec. 4.1 paragraphs 1-3). What’s more we followed the reviewer’s comments about robustness and added further experiments with noise, demonstrating that our representations compare favorably to a LASSO baseline over a wide range of noise types and magnitudes (Sec. 4.1, Fig. 3), including the trajectory-based noise suggested by the reviewer.
>
> To address the reviewer's remark about the limitations of the dictionary of functions used to approximate dynamics, we have adjusted the text (Sec. 4.1, paragraph 2) to emphasize the fact that one system, the simple oscillator (described in A.2.1), cannot in fact be expressed in the output basis since it has no closed polynomial form. Nevertheless, the system was where phase2vec had its largest performance advantage over the LASSO baseline. In general, the polynomial dictionary of cubic degree was shown to be sufficient for dynamical systems (Iben and Wagner 2021), similar to dictionaries in similar works, e.g. of(Champion et al. 2019), and we found it sufficient for the dynamics we sought to fit. However, we do consider the use of other dictionaries an important area of future work and have noted this in the Conclusion, paragraph 3.
>
> The reviewer’s note about the use of two-dimensional systems is well-received. We have adjusted the text (Introduction, paragraph 3) to motivate our focus on these systems. We would argue that the focus on two-dimensional systems is warranted for the first application of deep learning in phase space because of the importance of these systems in nature. Our climate demonstration, for instance, represents a critical application to two-dimensional systems. We do believe that there are extensions to other small systems (dimensions of 3,4,5, etc.), however, extensions to massive systems are less straightforward. The reviewer’s remarks have encouraged us to think of adaptations of our basic approach (feature learning in phase space) to these systems, as discussed in the Conclusion Section, paragraph 2.
> We believe the manuscript has greatly benefited from the reviewer’s remarks about the LASSO baseline for the auxiliary equation prediction task. As noted above, we have adjusted the training procedure so that phase2vec outperforms the LASSO baseline on both clean and noisy equation generalization tasks. The Results section has been updated to reflect this change (Fig. 3).
>
> Despite this improvement, we have emphasized that, indeed, the equation generalization task is ancillary to the later classification results (Sec. 4.1, paragraph 1). Moreover, we have extended our results to include a LASSO baseline on our denoising experiments (Sec. 4.2).
> Further, based on your remarks, we have introduced a fourth noise condition in which data are generated from binning velocities measured from trajectories whose initial conditions are chosen at random (Sec. 4.2, Fig. 3). We consider this a more realistic condition than the simpler noise regimes and we achieve as favorable results as in the other three noise conditions compared to a newly added LASSO baseline. Note that these new denoising experiments take steps towards answering the reviewer’s concerns about “completeness” of the dynamics map: now with relatively incomplete dynamics, including via the reviewer’s suggested trajectory method, we compare favorably to traditional methods.
>
> In future work, we hope to systematically explore the problem of generalization to new regimes of phase space, as mentioned in your final remark. We report anecdotally that, when the input and output phase volumes differ by a translation during the training period, phase2vec learns to approximate the input data in the new coordinate frame. However, whether or not this holds for testing data remains for future work, as we discuss in Conclusion, paragraph 3.
> Finally, we note that we have updated language and notation overall to answer the reviewer’s concerns about clarity, added the requested description of the model to the Appendix A.1, and addressed the minor wording-related notes (e.g. the spelling of "location" in Sec. 3.1). We will also release a python package and github repository which reproduces our results.

---

> > ### Comment · Reviewer_STRm · 2022-11-22
> > **Answer**
> >
> > I thank the authors for their answers, but that unfortunately does not change my first opinion on the paper.
> >
> > The fact that the results change drastically after changing the learning rate is worrying. It seems to me that this hyper-parameter is one of the first to be tuned. I now wonder how much the model can be improved by properly exploring the hyper-parameters. I misunderstand the phrase "to evaluate the model on a larger training set". Does this mean that the evaluation set size has increased, and therefore that your model is better on average than LASSO over a very large number of experiments ? Or is the training set size that increased?
> >
> > Based on Table 1, I'm not sure I understand what you mean: LASSO outperforms Phase2Vec in 5 cases for reconstruction, including the simple oscillator case. It is quite surprising that the parameters are better estimated with Phase2vec but that the state phase is more accurate with LASSO.
> >
> > Moreover, the limitation to 2D systems does not seem to me sufficiently justified by "the importance of these systems in nature": dynamics of robots is frequently 3D, Lorenz system (which model atmospheric convection) is 3D, Navier-Stokes is 3D even on 2D setups (speed along two axes and pressure), etc...

---

> > > ### Author Response · Authors · 2022-11-29
> > > **post-rebuttal response**
> > >
> > > We thank the reviewer for the continued remarks and would like to provide some additional clarification as well as a link to code with our basic results. We will clarify that in fact the improved results are due to, in our view, the changed hyperparameters as well as more stable evaluation on a larger test set (we had a typo which we will fix in the final version - we meant to write a larger test set and not a larger training set). We also emphasize that these hyperparameters were not initially tuned since our focus was, as we have noted, not on the auxiliary task, but on the downstream tasks, on which we had already achieved satisfactory performance. Following the reviewers’ comments about performance on this task, we thought it best to tune the hyperparameters, which was easy to do and resulted in improved performance. Our code is now available: https://anonymous.4open.science/r/phase2vec-E5F1.
> > >
> > > We are not sure what the reviewer means by “state phase is more accurate with LASSO”, as there are cases where LASSO is better and cases where P2V has lower error. On average, P2V is better across test systems, which comprise a representative sampling of two-dimensional dynamics. Moreover, P2V outperforms LASSO on the comprehensive polynomial testing set which takes into account all possible two-dimensional dynamics with equations up to cubic degree.
> > >
> > > To address the reviewer’s concerns about the focus on two-dimensional dynamics, we will first say that we agree with the reviewer that, while these dynamics are important, they are certainly not the only or most important in any sense. Moreover, and as we wrote in the discussion, extensions to >2 dimensional systems are possible with higher-dimensional convolutions. We are interested in systematically exploring not only these additional low-dimensional systems but also in massive systems which may require a categorically different approach that is beyond the scope of the current work.

---

> > > > ### Author Response · Authors · 2022-12-09
> > > > **3d extension of phase2vec**
> > > >
> > > > Following your suggestion, we have updated our code to include the 3d case. The relevant notebook, [train_and_eval_3d.ipynb](https://anonymous.4open.science/r/phase2vec-E5F1/notebooks/train_and_eval_3d.ipynb), qualitatively and quantitatively replicates our results for the 2d case. We outperform a per-equation LASSO fitting baseline on both held-out 3d saddle-node dynamics and the Lorenz dynamics. The results can be reproduced by running the notebook.
> > > >
> > > > Please also see our elaborated response above.

---

### Official Review · Reviewer_sZrD · 2022-10-24

**Confidence:** 4
**Correctness:** 3
**Technical Novelty And Significance:** 4
**Empirical Novelty And Significance:** 3
**Recommendation:** 8

**Clarity, Quality, Novelty And Reproducibility:**

The paper is very well written, to my knowledge original, clear and concise. Appendices give enough information for reproducibilty.

**Strength And Weaknesses:**

Phase space is a traditional way to understand what is happening in dynamical systems. This is a combined in a nice way to a traditional deep learning image classification pipeline, using the strengths of both the physical domain and the machine learning domain.  The idea of using equation terms as "words" in the encoding is excellent, as the decoding through solving the resulting sparse differential equation has a real chance of being the optimally sparse representation of the system.

Here lies also the weakness of the system - as small changes to a differential equation terms, can lead to big changes in the phase space, so the encoder may not be very stable. Restraining to polynomial functions of low degree may help on that, but at the same time they system becomes less able to encode such complicated behaviour. Would there be some Lipschitz condition based reasoning that would provide smoothness of the phase space transformation with respect to the changes of polynomial weightings or is the de-noising training with digital ensembles able to find stable encodings and move away for highly sensitive combinations? Or are the described normalization procedures able o solve this?

**Summary Of The Paper:**

The manuscript discuss how to encode 2D phase space images to shorter vectors. The embedding can be decoded to reveal the governing equations, recover simple, sparse models or de-noise the input preserving the dynamical relations. As the phase portrait is so fundamental in describing the dynamics, it is not a surprise the physical features like energy conversation, etc can be decoded out, as there are very visible in the phase space image.

The proposed pipeline feeds the phase portrait, through a bottleneck, to polynomials that can be used to reconstruct the phase plot through solving the decoded equation. A consistency error is set between the original and newly generated phase plots and the sparsity loss is set to keep the equations as simple as possible.

The training data is an ensemble of polynomial ODEs in such a way that the form of equation where the model was tested were removed from the training data - to check the generalisation capability. The results were compared to a traditional per-equation fitting method.

The authors demonstrate the encodings with an analysis of atmosphere.

**Summary Of The Review:**

The paper is valuable contribution to explainability research, where experimental data can be encoded in a physically meaningful manner that supports common intuition.  The network architecture is a variation of an auto-encoder, where part of the generator is replaced with a world model. I would really be interested seeing this connected to a control problem, where the intuition gained in the encoding is used to control the physical processes....  So, this is not only about seen that, done that, but also opens new ways for machine learning based, explainable, control systems, besides the more straightforward venue to dimensions greater than two.

---

> ### Author Response · Authors · 2022-11-18
> **Response to reviewer sZrD**
>
> The reviewer’s remarks about stability are well-received and especially important in a dynamical systems context. However, we believe that embeddings learned by phase2vec are sufficiently stable for one key reason. Namely, our denoising experiments, including ones in which parameters are perturbed, resulting in accurate reconstructions, indicating that small changes in the learned latent space have sensible effects in phase space (see Sec. 4.2). We have now included this argument in the Conclusion section.
>
> Many dynamical systems undergo violent changes in the topology of trajectories (bifurcations) when parameters are changed by a small amount. Indeed, most of our models (excluding the Lotka-Volterra model and the VanDerPol oscillator) are split into two classes depending on if their parameters are on one or the other side of a bifurcation boundary. Nevertheless, we never observed a comparatively large change in the embedding space as input systems underwent these shifts. Future work inspired by your remarks will investigate ways of incorporating sensitivity tests to these critical phenomena. Indeed, the tools the reviewer has suggested, for instance, the measurement of Lipschitz constants of the embedding network will be important to this future work. We now address this in the Conclusion section, paragraph 3.
>
> We are very intrigued by the reviewer’s remark about control applications and consider it an interesting parallel to our mention of “design”. We imagine that, if embeddings could be demonstrated to encode the probability of exhibiting a certain target behavior, phase2vec could be used to “steer” input dynamics. An interesting challenge will be to apply phase2vec to input trajectory data, the common setting for control applications.  We have noted this potential application in the final sentence of the Conclusion.

---

### Official Review · Reviewer_HbBX · 2022-11-02

**Confidence:** 3
**Correctness:** 3
**Technical Novelty And Significance:** 3
**Empirical Novelty And Significance:** 3
**Recommendation:** 8

**Clarity, Quality, Novelty And Reproducibility:**

**Clarity and quality:** the paper is generally well written and well motivated. The presentation is of high quality.

**Novelty:** the main technical contribution of phase2vec - combining a 2D ConvNet and self-supervised loss applied to dynamical systems - is novel to the best of my knowledge.

**Reproducibility:** No code is provided, which can limit follow-up work. However, datasets and experiments are described in some detail in the appendix.

As previously stated, I will be monitoring other reviewers' opinion on the novelty of the technical contributions as well as the experimental rigour.

**Strength And Weaknesses:**

### Strengths
+ The paper is generally well written and well motivated. The Introduction is readable by non-experts and contextualises the contributions well.
+ The proposed technique is straightforward and easy to understand. The main technical contribution - combining a 2D ConvNet and self-supervised loss applied to dynamical systems - is novel to the best of my knowledge. Phase2vec is an autoencoder for dynamical systems - which by itself isn't new - but the formulation of using a 2D ConvNet and self supervision signal is. (I would like to see the authors address the question of novelty during the rebuttal, see my questions below.)
+ The experimental evaluation seems thorough - phase2vec is probed for its ability to generalise outside its training distribution as well as on real world data. (But I would be keen to read other reviewers' opinion on the novelty of the technical contributions as well as the experimental rigour.)

### Weaknesses
- As the entire methodology and experiments are only geared towards dynamical systems in 2D Euclidean space, I would have liked to see this mentioned in the Abstract/Introduction. The only place where this is mentioned is in the Conclusion. (I realise this can be opinionated.)
- I found the Methodology and Related Work sections to have minor clarity issues which could be addressed by re-writing. I would like the authors to better contextualise the novelty of the proposed approach.

### Questions and Feedback
1. I appreciate the Related Work headers. While the section does a good job at surveying the literature, I was unsure how this paper, its methodology, and its experiments fit into each of those headers. Could the authors contextualise how their work is situated compared to what has been done. This may help to explicitly identify the main novelty of the paper w.r.t. existing literature. As a concrete example, there is a paragraph on vector-field-based approaches - but does phase2vec use them - if yes, how?
2. Along this line of contextualising the novelty w.r.t. related work, could the authors clarify whether the proposed self supervised reconstruction loss is original? Are there similar works which encode observations to a latent dimension and run dynamics there?
3. Section 3.1. describes the 2 x n x n input array. How is the size n chosen? Does n need to be fixed? Assuming not, could the authors comment on how sensitive phase2vec is to changes in the input size?
5. Currently, the 2D ConvNet operates on a 2D grid array where each pixel/entry is a 2D velocity vector. Has phase2vec made any assumptions about the symmetries that may act on this vector space? E.g. is it realistic to assume that the velocity vectors should be rotationally equivariant (i.e. if the system is rotated, they would rotate along with it)? How would additional symmetry constraints impact the design of phase2vec and the underlying ConvNet + self-supervised loss function used?

### Minor Comments
- Along these lines, in Figure 1, it may be useful to list the dimensionality of the intermediate tensors. Just to double check my understanding: the input is 2 x n x n, it goes through a series of convolutions, the final prediction is 2 x 10?
- I found the paragraph under equation 2 to be unclear. I assume it describes the basis functions? What is the multi-index? Why cubic degree? Would the resulting basis be: $1, x_1, x_2, x_1^2, x_2^2, x_1 x_2, x_1^2 x_2, x_1 x_2^2, x_1^3, x_2^3$?

**Summary Of The Paper:**

This paper proposes phase2vec - a self-supervised framework for learning representations of dynamical systems in 2D. The main technical contribution is to use a 2D convolutional network for embedding dynamical system velocities, and formulating a reconstruction loss based on the data creation process to train the network weights. Experiments demonstrate that phase2vec can generalise to previously unseen dynamics equations, noisy systems, and real-world climate data, as well as serve as a downstream classifier for physical properties of dynamical systems.

**Summary Of The Review:**

I currently recommend acceptance. Overall, the paper is well written, the proposed phase2vec method is straightforward, and the experimental evaluation demonstrates its usefulness across several scenarios. I would like the authors to better contextualise the novelty of the proposed approach in the rebuttal.

---

**Post-rebuttal:** Thank you to the authors for their rebuttal and revised manuscript. Based on the rebuttal as well as the other reviewers' summary, I would currently like to keep my original score/assessment.

**Post-discussion:** Apologies for the late action. Based on revisiting the discussion and the authors releasing open source code (I have not tried running it, but it looks modular, well thought out, and extensible), I have upgraded my score. I think the revised manuscript is an improvement over the original and addresses reviewers' concerns. I would like to re-iterate my moderate confidence.

---

> ### Author Response · Authors · 2022-11-18
> **Response to reviewer HbBX**
>
> We have made numerous changes in response to reviewer 2’s remarks and we believe the manuscript is significantly improved and clarified as a result. Importantly, we have adjusted the abstract and introduction (paragraph 3) to indicate that our focus is two-dimensional systems, in response to the reviewer’s first bullet-pointed weakness. We believe our focus in this respect is justified because of the preponderance and importance of two-dimensional systems in nature and have updated the text accordingly. Nevertheless, extensions to higher and even (very high) dimensions is an exciting area of future work which we discuss in the Conclusion Section, paragraph 2.
>
> To address the reviewer’s second weakness bullet point, we have made numerous clarifications throughout the text and better contextualized the novelty of our work. Specifically, we have added to each paragraph of the related work section some new remarks which explain how our approach differs from earlier methods (e.g. in paragraph “Generalization of physical systems”: “[Manifold boundary approximation] can be used to identify dynamics that happen to lie on the boundaries of the model manifold, but its use in identifying arbitrary, user-defined dynamical classes is less straightforward.”)
>
> To answer the reviewer’s remark specifically about vector-field-based approaches, we have updated the corresponding paragraph in Related Work. Our self-supervised loss shares features with earlier work but extends it in important ways. For example, (Ye et al. 2020) used an autoencoder to extract features from a vector field of air flow, but did this only for one type of system in order to address a very specific, supervised problem of predicting air pressure on an immersed cylinder. The authors did not learn features of general physical relevance or use anything like a physics-informed loss. To answer the reviewer’s remark about the context of our work in existing literature, there are important examples of papers in which dynamics are run in a latent space (Lusch et al. 2018, Champion et al., 2019), but these papers neither use vector fields nor try to learn features for more than one system at a time (which is what the use of vector fields enables). We are not aware of other works using the fixed-point normalization we use, however, the importance of dynamical slow-down near bifurcation points has been recognized previously (Tredicce et al. 2004).
>
> The reviewer also remarked on our choice of n, the effective resolution of the phase space data. We had experimented earlier with both smaller and larger values for n, finding no effect on reconstruction quality, though it is reasonable to assume that for extremely small values feature quality will degrade  We have clarified this point in the text (Sec. 3.1 paragraph 1), and included in the supplement an example of n=32, 64 and 128 (Fig. A.15).
>
> We thank the reviewer for the insightful remarks about symmetry. We have added a new paragraph (Conclusion paragraph 3) to include the few assumptions and symmetries our model relies upon. The most important among these arises from the use of convolutional filters, which add a degree of translation covariance to our features, a desired property when modeling  phase space data (and not a limiting assumption). Convolutional networks are known to exhibit a small degree of rotation covariance, so finding systems where this symmetry is present is an interesting area of future work. In general, the exploration of new symmetries and physics-informed assumptions is a major research avenue for phase2vec, as we now discuss in the Conclusion Section, paragraph 3.
>
> Finally, to address the reviewer’s minor comments, we have included additional architectural details, including the size of intermediate tensors, in Sec. 3.1, paragraph 2 and Sec. A.1, paragraph 1. Yes, the final output is 10 x 2, which is produced first by a series of convolutions and ultimately by a fully-connected decoder. The notation of Sec. 3.1 paragraph 2, has also been clarified in general, the multi-index notation has been removed, and concrete examples have been added.

---

> > ### Comment · Reviewer_HbBX · 2022-11-22
> > **Post-rebuttal**
> >
> > Thank you to the authors for their rebuttal and revised manuscript. Based on the rebuttal as well as the other reviewers' summary, I would currently like to keep my original score/assessment.

---

> > > ### Author Response · Authors · 2022-11-29
> > > **post-rebuttal**
> > >
> > > Following our previous response to the reviewer and the updates we have included here, we are keen to address additional concerns the reviewer still holds.

---

> > > > ### Comment · Reviewer_HbBX · 2022-12-07
> > > > **Post-discussion**
> > > >
> > > > Apologies for the late action. Based on revisiting the discussion and the authors releasing open source code (I have not tried running it, but it looks modular, well thought out, and extensible), I have upgraded my score. I think the revised manuscript is an improvement over the original and addresses reviewers' concerns. I would like to re-iterate my moderate confidence.

---

### Official Review · Reviewer_PePd · 2022-11-04

**Confidence:** 3
**Correctness:** 3
**Technical Novelty And Significance:** 2
**Empirical Novelty And Significance:** 3
**Recommendation:** 8

**Clarity, Quality, Novelty And Reproducibility:**

Although I am not familiar with literature on the specific challenge addressed by authors, I do believe they did a good job with their related work section and showing the novelties of their approach. This paper proposes a good method, but it is hard to follow on its most technical parts, and its evaluation does not seem convincing (for the reasons outlined above).

**Strength And Weaknesses:**

Strengths:
- S1. Interesting and important problem.
- S2. Recapitulation of relevant related work is very good.
- S3. The methodology is sound and non-trivial contribution.

Weaknesses:
- W1. The evaluation does not appear to be systematic. In neither the paper nor the appendix, the authors specify which dataset/ODE they are using for the examples of dynamically-informed denoising. It is possible that this is anecdotal evidence and not a legitimate claim for the paper. Additionally, I wonder why authors did not compare their method with other denoising methods and compute error metrics.
- W2. I found their method's design choices difficult to understand. There were some of them that I had to read several times from the appendix to the main paper. Other times, the information was simply not available. For example, why should we use the cubic expansion? Did you find this out experimentally?
- W3. The physics-informed term commonly refers to the physics-informed neural networks (PINNs), but authors use it with a significantly different meaning. It might be a good idea to think of another term or at least another title.
- W4. Readability would be benefited from putting clear examples, especially when addressing the mathematics in Section 3.

Questions:

- Q1. Can you please response to questions in W1?
- Q2. Please see questions stated in W2.
- Q3. Can you elaborate on why LASSO should be the only baseline for results in generalization to held-out systems? It is hard to judge if these are good results when compared to one simple baseline, and it’s not clear which were the reasons to exclude them from the evaluation to the several methods listed in the related work.
- Q4. From which literature did you obtain the physical properties of the underlying data (stability of fixed points, conservation of energy, and the incompressibility of flows)? Is this an exhaustive list?


**Summary Of The Paper:**

This paper introduces a methodology for predicting the equations of polynomial ODEs and obtaining embeddings of dynamical systems which can be applied to denoising data and classifying time series based on their physical properties. The authors propose a novel architecture based on CNNs to encode geometric features of vector fields, which is trained to minimize reconstruction losses. A FC layer predicts polynomial ODE coefficients, which are used for predicting equations that haven't been seen previously. They present results in synthetic datasets generated from polynomial ODEs, and utilize meteorological data to showcase their embedding space is meaningful.

**Summary Of The Review:**

The main concern I have with this paper is the lack of systematic evaluation in some sections. Although I have some concerns about the presentation, I hope authors add more details and make it more digestible. In my opinion, this paper is marginally over the threshold for acceptance. I would be willing to increase my score with the authors responding to my concerns on evaluation.

----
**After authors response:** I changed my decision to ACCEPT. See thread for details.

---

> ### Author Response · Authors · 2022-11-18
> **Response to reviewer PePD**
>
> We thank the reviewer for these important comments, and we believe the changes to reflect these comments greatly strengthen the paper. Most importantly, we adjusted our training procedure (lower learning rate of 1e-4 over a longer training period of 200 iterations) in order to out-compete the LASSO baseline on the equation generalization task. Although we still consider this task ancillary to our later demonstrations, we believe this improved performance strengthens our argument.
>
> To address point W1, we have clarified in Sec. 4.2 paragraph 1 that dynamically informed denoising is carried out on the test set of “classical” systems, detailed in Sec. A.2.1. We have also updated the text to emphasize that, while denoising is an interesting application of phase2vec, its main use in the current demonstration is to show that embeddings are stable enough to be used for downstream classification problems, including on real, potentially noisy data. Nevertheless, we have additionally updated Fig. 3 to quantitatively depict the relationship between noise magnitude and reconstruction error, which compares favorably to a LASSO baseline (with example in Figures A.11-14). We find that noise can significantly perturb input data without incurring a large reconstruction penalty (Fig. 3). While there are other potential baselines for both the clean and noisy generalization tasks, we feel justified in including only LASSO, the simplest possible competitor, since our current interest is in the downstream use of embeddings for classification. Now that phase2vec has been demonstrated to perform favorably compared to this baseline, we look forward to developing the equation prediction and denoising capabilities of the model in future work.
>
> To address point W2, we have endeavored to clarify the text overall, but especially Sec. 3, where we now explain in greater detail our choice of hyperparameters. To address the reviewer’s specific remark, we have edited the text (Sec. 3.1 paragraph 2) to explain that our choice of a cubic dictionary is based on earlier work (Iben and Wagner 2021) which showed this value was sufficient for a wide array of non-linear systems. While similar earlier works like (Brunton et al. 2016) have used greater degrees (c=5), we would argue that increasing c is not crucial since, after all, phase2vec can already fit data which has no representation as a finite sum of polynomial terms (e.g. the simple oscillator class described in Appendix A.2.1 and Table 1).
>
> We agree with the reviewer in the context of point W3 that the term “physics-informed” most often refers to a machine learning framework incorporating a physics-informed neural network (PINN) where a specific equation is used “in the loop”. However, we feel justified in applying this term to our setting since phase2vec is biased to learn representations which have physical meaning, via a physics-informed normalization. We believe this places phase2vec squarely under the “learning biases” heading in the physics-informed community, as outlined in (Karniadakis et al. 2021), Box 2. We have updated the text in two places (Introduction, paragraph 3; Sec 3.2, paragraph 1) to reflect this.
>
> Next, we have justified our use of the abstract physical classes analyzed in the section “Decoding the physics of embedded data”. We have updated the text to explain how linear stability vs the compressibility/conservativity tasks are prime examples of local and global properties of vector fields, respectively. Further, we have noted that Helmholtz’s theorem implies that, under mild assumptions, all vector fields are a sum of an incompressible and a conservative components (Strogatz et al. 1994). So, in a sense, these properties are the two basic building blocks of all realistic vector fields.
>
> To further clarify our method, we have included new, concrete examples in line with remark W4. In Sec 3, paragraph 1, we have contextualized the mathematics with a specific dynamical model, the homoclinic bifurcation family, so that the readers ground the notation in an actual system analyzed later in the text. Further, to visualize the effect of our physics-informed loss, we have updated the appendix Sec. A.1.1 to include a figure showing how fixed-point normalization emphasizes slow regions in the dynamics over and above a vanilla euclidean loss (Fig. A.7).

---

> > ### Comment · Reviewer_PePd · 2022-12-11
> > **Response to authors**
> >
> > My apologies for the late response.
> >
> > I thank the authors for the changes they made and for addressing the insightful comments/concerns of other reviewers. I found the updated version of the paper easier to understand.
> >
> > I especially appreciate the updates to Fig. 3. Instead of the anecdotal plots from before, the evaluation is now conducted systematically. And thanks for explaining the choice of the baseline for Fig. 3. According to my understanding, generalization and reconstruction experiments are sanity checks (auxiliary tasks) and decoding/classification results are the main focus of the paper. For sanity checks, it makes sense not to compare all baselines, but for the main results, you must compare all baselines. In fact, you did it, but your thought process wasn't clearly stated (at least I couldn't find it). So, I would advise authors to clearly state which are the main results and which are preliminary checks in the camera-ready version. Perhaps you can add this at the top of section 4.
> >
> > Following the resolution of my concerns, I changed my decision to ACCEPT.

---

### Author Response · Authors · 2022-11-18
**OVERALL REMARKS**

We would like to thank all of the reviewers for their comments, and we believe the changes we have consequently made to the manuscript improve it significantly. Following the reviewers' remarks we have made extensive changes to the text to improve the clarity, readability, and consistency of the notation. The Related Work section has been extensively updated to compare and contrast existing techniques with our method. Further, new details about the architecture and training procedure were added to the Methodology and Appendix section A.1, as well as figures which help depict the effect of fixed-point normalization and the resolution of input data. We are very intrigued, nevertheless, about extensions of our work in this direction in future work.

Here, we would like to emphasize that in the current setting, we consider the classification of physical properties as the principal task, so as we have made an attempt to clarify in the text, comparison to other baselines on the auxiliary task is not our main focus. With that, following the reviewers' remarks, we have improved phase2vec performance (by training with a lower learning rate compared to a LASSO baseline on both clean and noisy equation generalization tasks.

At last, upon publication, we will release a python package and github repository for reproducibility.

---

> ### Author Response · Authors · 2022-11-29
> **code availability**
>
> Our code is now available: https://anonymous.4open.science/r/phase2vec-E5F1

---

> > ### Author Response · Authors · 2022-12-09
> > **3d extension of phase2vec**
> >
> > In response to the reviewer's remarks, we have updated the above code to include a 3d extension of `phase2vec`.
> > The relevant notebook, [train_and_eval_3d.ipynb](https://anonymous.4open.science/r/phase2vec-E5F1/notebooks/train_and_eval_3d.ipynb), qualitatively and quantitatively replicates our results for the 2d case.
> >
> > Quantitative results are included in the notebook and can be reproduced by running the notebook cell by cell. In short, we outperform a per-equation LASSO fitting baseline on both held-out 3d saddle-node dynamics and the Lorenz dynamics suggested by reviewer STRm.
> >
> > While this result confirms our previous assertion that `phase2vec` scales easily to n-dimensional dynamics for small n and answers reviewer remarks, we still maintain that the principle challenge for future work is the extension to the large n case, where high-dimensional convolutional features are more difficult to learn

---

### Decision · Program_Chairs · 2023-01-20

**Decision:**

Accept: notable-top-25%

**Justification For Why Not Higher Score:**

Method is only applicable to 2D systems

**Justification For Why Not Lower Score:**

I found this to be an interesting and innovative paper. Reviewers mostly agree it is of high quality (8,8,8,5), with the concerns raised by (5) seemingly minor

**Metareview: Summary, Strengths And Weaknesses:**

The paper introduces a new method for embedding dynamical systems and predicting ODE equations, which can be applied to denoising and classification. Reviewers agree that this is an interesting and important problem, and that the method is sound and well-motivated. Most reviewers agreed that the experiments are thorough and show promising results. The primary concern, raised by multiple reviewers, is that the method is only applied to 2D systems, and can only ever work for rather low-dimensional systems in its current form. In my view the authors have done a decent job motivating the importance of 2D systems, and have provided a 3D implementation as well. As such I recommend the paper be accepted to ICLR.

**Note From Pc:**

if the above contains the word "oral" or "spotlight" please see: "oral" presentation means -> notable-top-5% and "spotlight" means -> notable-top-25%. As stated in our emails, we are disassociating presentation type from AC recommendations